# Immersive Visualization of the Classical Non-Euclidean Spaces using Real-Time Ray Tracing in VR

Luiz Velho, Vinicius da Silva, and Tiago Novello

Instituto de Matemática Pura e Aplicada

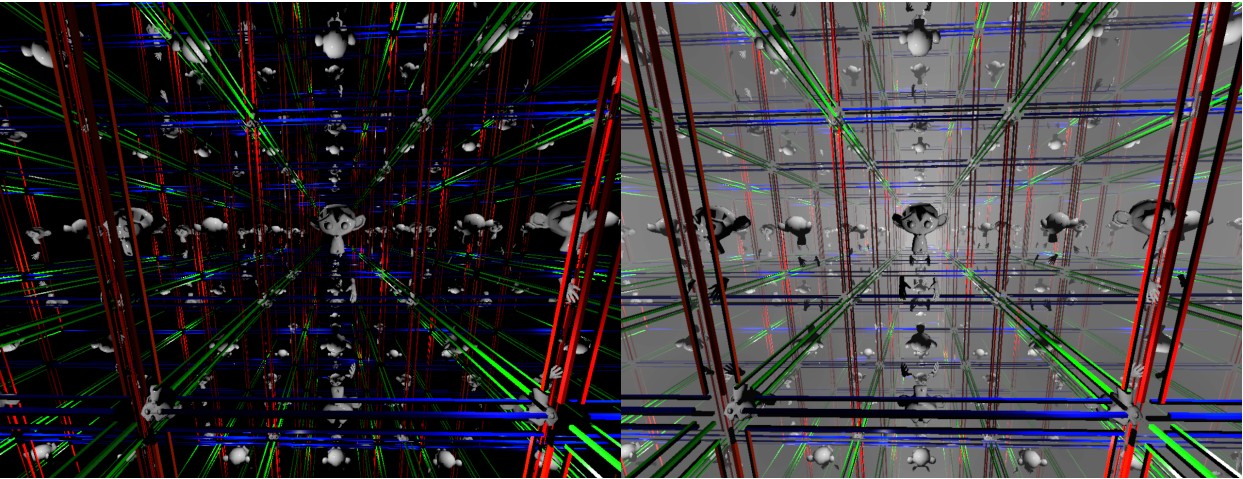

Fig. 1. Inside view of a three-dimensional torus. The right image includes the simulation of fog to better convey the sense of depth.

**Abstract**—This paper presents a system for immersive visualization of the Classical Non-Euclidean spaces using real-time ray tracing. It exploits the capabilities of the latest generation of GPU's based on the NVIDIA's Turing architecture in order to develop new methods for intuitive exploration of landscapes featuring non-trivial geometry and topology in virtual reality.

**Index Terms**—Ray tracing, VR, Non-Euclidean geometry

## 1 INTRODUCTION

In late 2018, NVIDIA introduced a new generation of GPUs that according to Jensen Huang, the company's CEO, is a major breakthrough in the history of the computer graphics industry. The Turing architecture of the RTX GPUs was developed over the past 15 years to make possible the implementation of ray tracing algorithms in real-time, thus enabling visualization applications with an unprecedented degree of photo-realism.

In this paper, we take the challenge of applying the power of this new generation of RTX GPUs in the exploration of mathematical spaces that feature non-trivial geometry and topology in virtual reality.

### 1.1 Motivation

The Turing architecture combines traditional capabilities of the previous generations GPUs for *rasterization* (Graphics Pipeline) and *compute* (CUDA) with new capabilities for *artificial intelligence* (Tensor Core) and *ray tracing* (RT Core). Overall, these features in aggregate form a powerful set of complementary resources for the development of new media applications, not possible until now.

While the most obvious use of the RTX GPUs is for real-time photo-realistic simulation with applications in entertainment, architecture, design, etc., there are other areas where this power can open up new perspectives not imaginable before. One of these areas is the *Visualization of Mathematics*. In this realm, abstract concepts, such as: high dimensional spaces; Non-Euclidean geometries; non-trivial topologies and manifolds, can be made concrete for immersive exploration.

Using Virtual Reality and Ray Tracing it is now possible to create these Mathematical landscapes for interactive visualization, literally putting the viewer inside these abstract worlds for an intuitive understanding. Such experiences have the potential to allow many insights with great impact in research and education, among other aspects.

### 1.2 Contributions

The main contribution of our work is the development of an experimental platform for the immersive visualization of Non-Euclidean Spaces using real-time ray tracing. This includes the design and implementation of an extensive framework for creating interactive experiences in landscapes that can model different types of three-dimensional manifolds/orbifolds.

In addition, in order to test our platform and validate its effectiveness, we produced a series of virtual reality applications and conducted informal user studies that give directions for future research.

The system is implemented on top of NVIDIA's Falcor [2] real-time rendering framework using DirectX 12 (DXR) on Windows 10. For that, we relied on a Falcor extension in order to integrate Ray Tracing with Virtual Reality [13]. To the best of our knowledge, this is the first project that uses RTX combining real-time ray tracing and virtual reality for the exploration of abstract mathematical landscapes.

### 1.3 Structure of the Paper

The paper is structured as follows: Section 2 reviews previous and related work; Section 3 introduces the basic mathematical concepts associated with our work; Section 4 presents the method for GPU ray

---

- *Luiz Velho is with IMPA. E-mail: lvelho@impa.br.*
- *Vinicius da Silva is with IMPA. E-mail: dsilva.vinicius@gmail.com.*
- *Tiago Novello is with IMPA. E-mail: tiago.novello90@gmail.com.*

Graphics Interface Conference 2020
28-29 May

tracing of 3D Manifolds/Orbifolds; Section 5 shows examples of experiments and discusses the analysis of our results; Section 6 presents a quantitative and qualitative analysis of our results; Section 7 elaborates on possible extensions of our platform and suggests perspectives for future work; finally Section 8 provides concluding remarks.

## 2 RELATED AND PREVIOUS WORK

In this section we review previous work for visualization of Non-Euclidean spaces and report on other related work that are relevant to our research.

### 2.1 OpenGL Visualization of Non-Euclidean Spaces

Historically, the main effort for mathematics visualization, particularly of Non-Euclidean Spaces, took place at the Geometry Center during the period of 1994 to 1998. This initiative, under the leadership of William Thurston, resulted in a scientific program to study and disseminate modern geometry using interactive visualization.

Since Thurston's personal research focused primarily on hyperbolic manifolds [14], it was natural that the Geometry Center investigated the visualization of manifolds and orbifolds. For this purpose, a platform called *Geomview* [1] was developed. The software was based on OpenGL and supported interactive viewing in Euclidean, elliptical and hyperbolic geometries. Geomview featured a plugin architecture that made possible, among other things, the development of a module for the visualization of manifolds [7].

### 2.2 Virtual Reality

Researchers at the Geometry Center, already at that time realized the potential of Virtual Reality for providing insights into the world of geometric structures. Therefore, they created simple VR installations to allow the user, not only to have a glimpse at the visual landscape inside a 3-manifold, but also to experience the sensation of being immersed in such an environment. Two of their projects are Mathenautics [10] and Alice [5].

Another initiative in the direction of using virtual reality for mathematics visualization was JReality [16], a Java based 3D scene graph package designed for mathematical visualization at TU-Berlin. It can be used for creating immersive views of 3-manifold and relies on JOGL as a back-end for interactive OpenGL rendering.

### 2.3 Ray-Tracing

The early work on interactive visualization of Non-Euclidean spaces, reported above, was based on the traditional OpenGL rasterization pipeline. Therefore, the rendering algorithms employed a *Scene-Based* architecture.

The first work to propose the use of an *Image-Based* architecture for visualization of Non-Euclidean spaces using GPUs was from Berger et al. [3]. Their rendering algorithm exploited programmable compute shaders and CUDA to implement ray-tracing on the GPU. We take this seminal work to the next level, exploiting the capabilities of the latest graphics technology to implement an extensible framework for the development of virtual-reality exploration of mathematical landscapes.

Currently, the NVIDIA's Falcor rendering framework [2] provides a platform based on Vulkan and DirectX 12 that supports many features for real-time visualization, including OpenVR and DirectX Raytracing. However, ray tracing and virtual reality do not work in an integrated way in Falcor. In this work, we integrate ray tracing and virtual reality in Falcor for enabling the intrinsic visualization of 3-manifolds.

### 2.4 Metric Neutral

One relevant aspect in the visualization of Non-Euclidean spaces is the metric implied by the geometry. In this respect, Gunn [8] proposed a metric-neutral framework that simplifies the rendering of such geometries. Particularly, it introduces some advances that have an impact on the architecture of generic Virtual Reality systems — for example, a metric-neutral algorithm for head-tracking in VR for the different metric spaces of interest.

Another approach that simplifies the implementation of rendering applications for Non-Euclidean geometries was proposed by Guimaraes et al. [6] for two-dimensional manifolds. It is an encapsulation method to dissociate the application development from the geometric space in which it will be represented, while at the same time preserving the intrinsic metric and topological structures of the space.

We extend the approach of Guimaraes et al. from 2D to 3D, in order to facilitate the development of applications in our framework.

## 3 NON-EUCLIDEAN SPACES

In this section we introduce the concepts and main results of *manifolds* and some special *non-manifolds*: *polyhedral complexes* and *orbifolds*. We also present the main ingredients for a ray tracing implementation on such abstract spaces.

### 3.1 Ray Tracing Requirements

The paper deals with an immersive visualization of spaces modelled by Non–Euclidean geometries using ray tracing, thus we need at least three properties:

- Being locally similar to an Euclidean space — that is, a *manifold*. This allows us to put the viewer and the scene inside the ambient as in common approaches: some deformation may be allowed;

- For each point $p$ we need vectors pointing in all directions: the *tangent vectors* at $p$. Moreover, the *inner product* between two tangent vector is required. These definitions are used to simulate effects produced between the lights and the scene objects.

- For a point $p$ and a vector $v$ tangent at $p$, we should be able to compute the *ray* leaving $p$ in the direction of $v$. Finally, the intersection between rays and the scene "objects" are required.

*Geometric manifolds* satisfies the above properties. Such objects are locally geometrical similar to special spaces called *model geometries*. In dimension two, for example, there are exactly three models: Euclidean, hyperbolic, and spherical spaces. In dimension three, there are five more model geometries, however, in this work we focus on the (classical) first three spaces. We describe these topics in more details below. Great texts on this subject are Thurston [14] and Martelli [11].

### 3.2 Geometric Models

The spaces presented in this section will be very useful to model more complexes spaces which we should introduce later. The main ingredients for a ray tracing implementation are also present here.

**Definition 1** (Euclidean space)**.** The *Euclidean space* $\mathbb{E}^3$ is $\mathbb{R}^3$ endowed with the classical *inner product* $\langle u, v \rangle_{\mathbb{E}} = u_x \cdot v_x + u_y \cdot v_y + u_z \cdot v_z$ where $u = (u_x, u_y, u_z)$ and $v = (v_x, v_y, v_z)$ are vectors in $\mathbb{R}^3$. The *distance* between two points $p$ and $q$ is defined by $d_{\mathbb{E}}(p,q) = \sqrt{\langle p-q, p-q \rangle_{\mathbb{E}}}$. The curve $\gamma(t) = p + t \cdot v$ describes a *ray* leaving a point $p$ in a direction $v$. Analogously, for any $n > 0$ the Euclidean space $\mathbb{E}^n$ is constructed.

**Definition 2** (Hyperbolic space)**.** The *Lorentzian inner product* of the vectors $v$ and $u$ in $\mathbb{R}^4$ is defined as $\langle u, v \rangle_{\mathbb{H}} = u_x v_x + u_y v_y + u_z v_z - u_w v_w$. The vector space $\mathbb{R}^4$ endowed with the Lorentzian product is called the *Lorentzian space*. The *hyperbolic space* $\mathbb{H}^3$ is the hyperboloid $\{p \in \mathbb{R}^4 | \langle p, p \rangle_{\mathbb{H}} = -1\}$ endowed with a special metric $d_{\mathbb{H}}(p,q) = \cosh^{-1}(-\langle p, q \rangle_{\mathbb{H}})$, where $p$ and $q$ are two points in $\mathbb{H}^3$. Due to its remarkable similarity to the sphere definition (see next definition), $\mathbb{H}^3$ is also known as *pseudo-sphere*.

A tangent vector $v$ to a point $p$ in $\mathbb{H}^3$ satisfies $\langle p, v \rangle_{\mathbb{H}} = 0$. Moreover, the *tangent space* $T_p \mathbb{H}^3$ coincides with the set $\{v \in \mathbb{R}^4 | \langle p, v \rangle_{\mathbb{H}} = 0\}$. The Lorentzian inner product is positive on each tangent space.

*Rays* in $\mathbb{H}^3$ are the intersections between $\mathbb{H}^3$ and the planes in $\mathbb{R}^4$ containing the origin. For instance, the ray leaving a point $p \in \mathbb{H}^3$ in a tangent direction $v$ is the intersection between $\mathbb{H}^3$ and the plane spanned by the vectors $v$ and $p$ in $\mathbb{E}^4$. Such ray can be parameterized as $r(t) = \cosh(t)p + \sinh(t)v$.

The space $\mathbb{H}^3$ does not contain any straight line, thus its rays can not be straight. However, it is possible to model $\mathbb{H}^3$ in the unit open ball

in $\mathbb{R}^3$ — known as *Klein model* $\mathbb{K}^3$ — such that in this model the rays are straight lines. More precisely, each point $p \in \mathbb{H}^3$ is projected in the space $\{(x,y,z,w) \in \mathbb{R}^4 | w = 1\}$ by considering $p/p_w$, the space $\mathbb{K}^3$ is obtained by forgetting the coordinate $w$.

The hyperbolic space is a model of a *Non-Euclidean* geometry, since it does not satisfy only the Parallel Postulate: given a ray $r$ and a point $p \notin r$, there is a unique ray parallel to $r$. For a ray $r$ in the hyperbolic space and a point $p \notin r$ there are an infinite number of rays going through $p$ which do not intersect $r$.

**Definition 3** (Elliptic Space). The 3-*sphere* $\mathbb{S}^3$ is the set $\{p \in \mathbb{E}^4 | \langle p,p \rangle_{\mathbb{E}} = 1\}$ endowed with the metric $d_{\mathbb{S}}(p,q) = \cos^{-1} \langle p,q \rangle_{\mathbb{E}}$.

As in the hyperbolic case, a tangent vector $v$ to a point in $\mathbb{S}^3$ satisfies $\langle p,v \rangle_{\mathbb{E}} = 0$. The *tangent space* $T_p\mathbb{S}^3$ corresponds to the set $\{v \in \mathbb{E}^4 | \langle p,v \rangle_{\mathbb{E}} = 0\}$. The space $T_p\mathbb{S}^3$ inherits the Euclidean inner product of $\mathbb{E}^4$.

A *ray* in $\mathbb{S}^3$ passing through a point $p$ in a tangent direction $v$ is the arc produced by the intersection between $\mathbb{S}^3$ and the plane spanned by $v$, $p$, and the origin of $\mathbb{E}^4$. Such ray can be parameterized as $r(t) = cos(t)p + sin(t)v$.

Again, the 3-sphere $\mathbb{S}^3$ is an example of a Non-Euclidean geometry, since its fails the Parallel Postulate: given a ray $r$ and a point $p \notin r$, there is a unique ray parallel to $r$. As the rays in $\mathbb{S}^3$ are the big circles, thus choosing two of then in $\mathbb{S}^2 \subset \mathbb{S}^3$, they always intersect in exactly two points.

### 3.3 Manifolds

A *n-manifold M* is a topological space which is locally identical (topologically speaking) to the Euclidean space $\mathbb{E}^n$; $n$ is the dimension of $M$. More precisely, there is a neighborhood of every point in $M$ mapped homeomorphically to the open ball of $\mathbb{E}^n$. These maps are called *charts* of $M$. The change of charts between two neighborhoods in $M$ must be continuous. Thus, informally, the manifold definition generalizes the concept of Euclidean spaces. This work focus on manifolds of dimension 3. Examples of 3-manifolds include the Euclidean, hyperbolic, and spherical spaces.

Straight lines are fundamental objects when working with ray tracing algorithms, since light travels along them. A manifold $M$ admits a generalization of such notion, the *geodesics*. To define them we need two additional tools. The first is the calculus framework, which is done by requesting changes of charts in $M$ to be diffeomorphisms — $M$ is called *differentiable*. This allows us to define for each point a *tangent space* and work with calculus on it. The second tool is the attribution of an appropriate metric on each tangent space — $M$ is called *Riemannian*. Then we can compute angles between *vectors* in tangent spaces (crucial in ray tracing), and distances between two points in $M$. Finally, a *geodesic* in $M$ is a curve such that locally it is the shortest path. We use the term *ray* instead of geodesics since the paper deals with ray tracing.

## 4 GPU RAY TRACING OF 3D MANIFOLDS

In this section we present the method for immersive visualization of 3D manifolds and orbifolds using ray tracing on the GPU. In this respect, we fully exploit the ray tracing capabilities of RTX platform to design a framework for exploration of Non-Euclidean spaces that is extensible and structured to handle effectively interactive application scenarios.

We will discuss first the basic principles of ray tracing in Non-Euclidean spaces, as well as, the general algorithm in CPU. Then, we will show how to map the computation to the RTX pipeline and present the details of GPU implementation.

### 4.1 Overview of the Method

The ray tracing algorithm is arguably the most natural method to produce visualizations of the intrinsic space of a 3D manifold/orbifold. Basically, it is necessary to adapt the traditional ray tracing of the Euclidean ambient space to take into account both the geometry and topology of the manifold/orbifold. The first aspect of this task is to simulate the ray path as it travels inside the space, starting from the point of observation until it intersects with a visible object. The second

aspect amounts to shading, that computes the illumination and evaluates the light scattered from the environment in the ray direction. Because of the non-trivial topology of the manifold/orbifold, the computation of the ray path requires tracking its orbit over the covering space – this is done by transporting the ray as it exits and enters the fundamental domain.

### 4.2 Algorithm in CPU

Let's study the basic ray tracing algorithm for polyhedral complexes that represent manifold/orbifold spaces – and compare it with the traditional ray tracing of Euclidean space, in order to understand the differences.

As it can be verified in Algorithm 1, the rays are generated from the observer's point of view (lines 1 - 3) and intersected with visible objects (line 5) and if there is a hit (line 6), shading is done (line 7).

These three steps are present in all ray tracing algorithms including the traditional one for the Euclidean space. In the case of ray tracing inside a manifold/orbifold we need extra steps to guide the path of a ray as it moves through the covering space. These correspond to lines 9, 10 and 12.

We assume that the whole computation has the *fundamental domain* as a base, which is modeled by a polyhedron $\Delta$. Therefore, as the ray hits a face $F_i$ in the boundary of the domain (line 9), we need to transport it by the action of the corresponding transformation of the discrete group (line 10).

For practical computational reasons we cannot continue the ray path indefinitely, thus a maximum level is set to stop the path (line 12).

Note that the most important and critical step is the group action (line 10), which is dependent of the geometry and topology of the manifold/orbifold. As such, it is specific for each type of space.

---

**Algorithm 1** Ray Tracing in manifolds/orbifolds

1: **for** each pixel $\sigma \in I$ **do**
2:     Let $p := 0$ and $v$ be the direction associated to $\sigma$
3:     Trace a ray $r$ from $(p,v)$ inside $\Delta$
4:     **repeat**
5:         Find closest intersection $i(r)$ with objects $O_0$ in $\Delta$
6:         **if** $i(r) \neq \emptyset$ **then**
7:             Shade pixel **break**
8:         **else**
9:             Find intersection of $r$ with faces $F_i$ of $\Delta$
10:            Compute the new origin $p'$ and ray $r'$.
11:         **end if**
12:     **until** $i \leq maxlevel$
13: **end for**

---

### 4.3 RTX Pipeline

NVidia RTX is a hardware and software platform with support for real time ray tracing. The ray tracing code of an application using this architecture consists of CPU host code, GPU device code, and the memory to transfer data to the Acceleration Structures for fast geometry culling when intersecting rays with scene objects.

Specifically, the CPU host code manages the memory flow between devices, sets up, controls and spawn GPU shaders and defines the Acceleration Structures.

The Acceleration Structure is divided conceptually in two classes: on one hand, the bottom level Acceleration Structure contains the rendering primitives (triangles for example); on the other hand, the top level Acceleration Structure is a hierarchical grouping of bottom level ones.

Finally, the GPU role is to run instances of the ray tracing shaders in parallel. This is analogous to the well-established GPU rasterization rendering pipeline.

However, despite the fact that the GPU rasterization pipeline is based on programmable shaders (i.e., vertex and pixel shaders), its structure was not designed for ray tracing. For this reason, before the introduction of RTX, the implementation of ray tracing algorithms in GPU using OpenGL was difficult and limited. Essentially, most of the

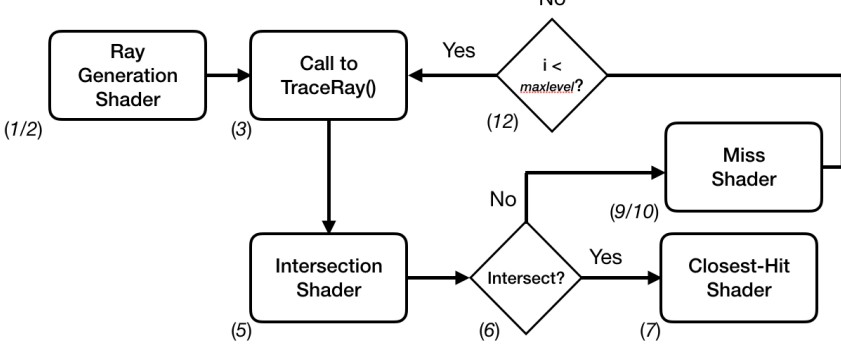

Fig. 2. Ray Tracing Pipeline - main stages of the RTX GPU computation flow (the numerical labels correspond to line numbers of Algorithm 1).

processing had to be done inside the pixel shader in a monolithic way, thus preventing scalable implementations using a modular architecture and also the definition of complex scenes.

In contrast, the ray tracing GPU device code runs under a pipeline scheme composed of a sequence of stages specifically designed for ray tracing operations. The goal of the first stages is to generate the rays. Afterwards, a fixed stage calculates the intersection of the rays with the scene geometry. Then, the intersection points are reported to the group of shading stages. Notice that more rays can be created at this point, resulting in a recursion in the pipeline. The final fixed stage outputs the generated image.

Each shader can be correlated with the tasks performed by the general CPU procedure described in Algorithm 1. The *Ray Generation Shader* is responsible for creating the rays (line 1), which are defined by their origins, directions and the custom user-defined data, called payloads (line 2). A call to TraceRay() launches a ray (line 3). The next stage is a fixed traversal of the Acceleration Structure which will describe only at high level here. This traversal uses an *Intersection Shader* to calculate the intersections (line 5). All hits found pass by tests to verify if they are the closest hit. After no additional hits are found, the *Closest-Hit Shader* is called for the closest intersection point (line 7). In case no hits are found, the Miss Shader is called as a fallback case. It is important to note that additional rays can be launched in the Closest-Hit and Miss shaders.

Figure 2 shows a simplified scheme of the pipeline, where the association of pipeline stages with the steps of the algorithm are indicated by the line numbers. More detailed information about RTX Ray Tracing can be seen in [17] and applications can be found in [9].

The above is the general ray tracing GPU pipeline. In the case of ray tracing inside a manifold/orbifold we have two classes of objects: i) the *scene objects* which are embedded in the space; and ii) the *boundary of the fundamental domain* that is represented by the polyhedron $\Delta$. They are treated differently when mapping the algorithm to the RTX pipeline — while the scene objects are tested and shaded in the regular way (lines 5 and 7), the boundary of the fundamental domain is used to transport the rays by the group action (lines 9 and 10). This is implemented with a custom designed Miss Shader.

Another important point is related to the Acceleration Structure. The RTX platform defines a hierarchical structure in order to efficiently guide the intersection of rays with scene objects. Bottom-level cells store the actual scene geometry while top-level cells hold pointers of the graph structure. In the diagram of Figure 2, this is encapsulated by the block for ray intersection (lines 5 and 6). Note however, that in the algorithm for visualization of manifolds/orbifolds, rays travel through the covering space entering and exiting the fundamental domain multiple times, In that respect, the fundamental domain acts as an special higher-level acceleration structure that defines the topology of the space.

The above description makes clear that the RTX platform potentially opens up new research directions for ray tracing applications, with an impact similar to the introduction of programmable shaders.

In particular, for the visualization of non-Euclidean spaces it allows non-trivial advances related to efficient and modular architectures for interactive and immersive exploration of scenes with complex geometry and topology, not possible until now.

### 4.4 GPU Implementation

The implementation of our visualization platform in GPU is build on top of Falcor using DirectX 12 on Windows 10. The Falcor development framework consists of a library with support for DXR at high level and a built-in scene description format.

We use the software Blender to create the scene objects and model the fundamental domains, including their boundaries.

The core functionality of our system's architecture consists of a set of shaders that are mapped to the RTX GPU pipeline as described above. In order to make the design of the system extensible and modular, we have adapted the metric neutral approach of Guimaraes et. al. [6] to ray tracing and extended it to 3D geometrical structures. In this context, we have developed generic shaders for each stage of the GPU ray tracing pipeline that are independent of the geometric structure of the manifold/orbifold. They are specialized and instanced based on the metric and topological properties of each individual space. That includes the model of the fundamental domain.

We now describe the tasks performed by the different shader classes, as well as, the mathematical operations necessary for the visualization of non-Euclidean spaces. Note that these operations are dependent of the Model Geometry being used in the space (i.e., Euclidean, Hyperbolic and Elliptic) as discussed in Section 3.

**Ray Generation Shader**: Creates camera rays. For this purpose it has to use the isometries of the space to transform the ray origin and direction to the camera coordinate system.

**Intersection Shader**: Computes the intersection between the ray and the scene objects. For this purpose it uses the parametric description of the ray. Both the ray and objects are defined according to the Model Geomery.

**Closest Hit Shader**: Performs the shading operation. This includes computing the local and global illumination. The local illumination amounts to direct contribution of light sources that is based on angles between the light direction and the surface normal, as well as, the distance to the light. All these operations are performed using the Model Geometry. Currently, we implemented only local illumination. Global illumination is a topic for future research as discussed in Section 7.

**Miss Shader**: Deals with the transport of rays in the covering space, as they leave and enter the fundamental domain. For this, the rays are tested for intersection with the boundary of the polyhedron $\Delta$. Here, both the geometric and topological aspects of the embedding space have to be taken into account.

The distinction of scene objects and the fundamental domain geometry is handled through a feature of Falcor's scene description, i.e., object and material ID's. These two types of entities have different ID's that causes the assignment of the appropriate specific shader classes. In this way, only objects in the scene are processed by the standard ray inter-

section operations, while the polyhedron representing the fundamental domain is processed only by ray-path propagation mechanism.

In addition, for the development of virtual reality applications, we employ and extend to Non-Euclidean spaces the Ray-VR algorithm [13] that implements stereo ray tracing on top of Falcor.

## 5 EXAMPLES AND RESULTS

In this section we present some expressive output images from our implementation of the algorithm in GPU using RTX, with examples of the classical manifolds and orbifolds.

Recall that manifolds are abstract spaces locally similar to the Euclidean space. We present three classical examples — Examples 1, 2, and 3 — of such spaces with their geometry modeled by the classical model geometries: Euclidean, hyperbolic, and spherical spaces.

The rays in such spaces have a particular behavior that can be explained in two ways. Topologically, these space are not simply connected: their fundamental group is nontrivial. Then by *Cartan's theorem* [4], there is a closed ray for each nontrivial element in the fundamental group. Algebraically, these spaces are the quotient of the model geometries by some discrete groups, producing thus a tessellation view inside the model geometry. These arguments explain the multiple copies of the scene in the examples below.

Orbifolds are modeled locally by quotients of a model geometry by discrete groups. Let $M$ be a Euclidean, hyperbolic, or spherical space. The quotient $M/\Gamma$ of $M$ by a discrete group acting on it could be a non-manifold. In this case, $M/\Gamma$ is called an *orbifold*.

We present two simple orbifold examples: the *mirrored cube*, and *mirrored dodecahedron* — Examples 4, 5.

### 5.1 Flat Torus

**Example 1** (Flat torus). Probably the most famous and easiest example of a compact 3-manifold is the *flat torus* $\mathbb{T}^3$. Topologically, it is obtained by gluing opposite faces of the unit cube $[0,1] \times [0,1] \times [0,1] \subset \mathbb{E}^3$. It is easy to check that the neighborhood of each point in $\mathbb{T}^3$ is a 3-ball of the Euclidean space. Thus $\mathbb{T}^3$ is indeed a 3-manifold.

$\mathbb{T}^3$ admits a geometric structure modeled by $\mathbb{E}^3$ since it is also the quotient of the Euclidean space by the group of translation spanned by $(x,y,z) \rightarrow (x \pm 1, y, z)$, $(x,y,z) \rightarrow (x, y \pm 1, z)$, and $(x,y,z) \rightarrow (x,y,z \pm 1)$. Thus, the face $[0,1] \times [0,1] \times 0$ is identified to $[0,1] \times [0,1] \times 1$ by the translation map $(x,y,z) \rightarrow (x,y,z+1)$. The remaining pairs of faces can be identified in an analogous way. The unit cube is the fundamental domain of $\mathbb{T}^3$.

A ray leaving a point $p \in \mathbb{T}^3$ in a direction $v$ is parameterized as $r(t) = p + t \cdot v$ in $\mathbb{E}^3$. For each intersection between $r$ and a face $F$ of the unit cube, we update $p$ by its correspondent point $p - n$ in the opposite face, where $n$ is the unit vector normal to $F$. The ray direction $v$ does not need to be updated.

Therefore, we have the ingredients for an immersive visualization of $\mathbb{T}^3$ using ray tracing. The scene can be set in the unit cube since it is the fundamental domain. The rays in $\mathbb{T}^3$ can return to the starting point, providing many copies of the scene. The immersive perception is $\mathbb{E}^3$ tessellated by unit cubes: each cube contains one copy of the scene.

Figure 3 provides an immersive visualization of the 3-dimensional torus $\mathbb{T}^3$, presented in Example 1, using the shader described in Subsection 4.4. There is only one monkey's head, the *Suzanne* classical Blender mesh, and a unique pair of hands. We attach Suzanne to the camera. The closed rays produce many scene copies. Algebraically, this image describes the action of the group of translation in the Euclidean space which covers $\mathbb{T}^3$, explaining thus the copies pattern.

### 5.2 Hyperbolic Dodecahedron

**Example 2** (Seifert-Weber dodecahedral space). To describe a compact 3-manifold with geometric structure modeled by the hyperbolic space consider a dodecahedron $P$. Identifying each pair of opposite faces in $P$ with an addition clockwise rotation of $3\pi/10$ gives rise to a manifold know as *Seifert–Weber dodecahedral space M*.

Face pairing produces many identifications, for example, you can verify that edges are grouped into six groups of five. Thus, it is not

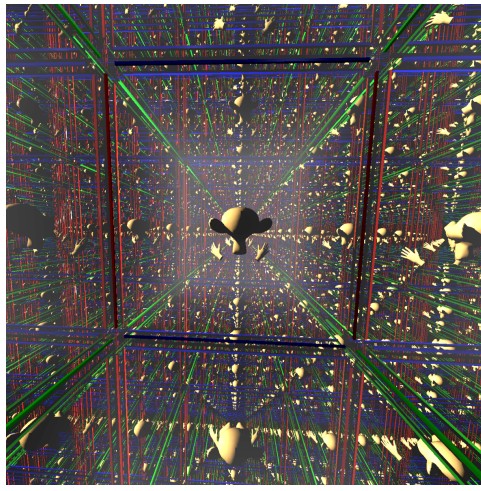

Fig. 3. Immersive view in the 3-dimensional flat torus. The space is obtained by identifying the opposite faces of a cube (fundamental domain). We use the cube to set up our scene: a unique mesh (Suzanne) endowed with hands, and the cube's edges (colored lines). The face pairing makes the rays that leave a face return from its opposite face, giving rise to many copies of the scene, tessellating the Euclidean space.

possible to fit Euclidean geometry into such a manifold, since the regular Euclidean dodecahedron has a dihedral angle of approximately 116 degrees. The desired dodecahedron should have a dihedral angle of 72 degrees.

We use the hyperbolic geometry to model the geometry of $M$. Let the dodecahedron be centered at the origin of $\mathbb{H}^3$. The dihedral angle of the dodecahedral in the hyperbolic space is smaller than in the Euclidean case. In fact, with an appropriate scale, the dodecahedron admits a dihedral angle of 72 degree as desired.

Using Klein's model of $\mathbb{H}^3$, the rays are straight. So to compute a ray leaving a point $p \in M$ in a direction $v$, we use $r(t) = p + tv$. For each intersection between $r$ and a dodecahedron face, we update $p$ and $v$ through the hyperbolic isometry that produces face pairing above. This isometry is quite distinct from Euclidean isometries (see [7]).

The immersive perception of $M$ using ray tracing is a tessellation of $\mathbb{H}^3$ by dodecahedra with a dihedral angle of 72 degrees.

Figure 4 illustrates an inside view of Seifert–Weber dodecahedral space, given in Example 2. Again, there is only one *Suzanne* endowed with hands attached to the camera. The image describes the action of a special discrete group (see Example 2) on the hyperbolic space, which provides a dodecahedron tessellation of the hyperbolic space.

### 5.3 Spherical Dodecahedron

**Example 3** (Poincaré dodecahedron space). If the opposite faces of the dodecahedron are identified by a clockwise rotation of only $\pi/5$ we get *Poincaré dodecahedron space*, a manifold discovered by Poincaré. This manifold is also known as *Poincaré homological sphere* since its first homological group is trivial.

Again, the face pairing forces many identifications. The edges are grouped into ten groups of three edges. To model the geometry of such space the dihedral angle of the dodecahedron must be 120. It is not possible to model with Euclidean geometry. In this case, we use spherical geometry.

To find the desired dodecahedron we consider it embedded in the 3-sphere. If the dodecahedron is very small its dihedral angle is very close to the Euclidean dodecahedron. Then, with an appropriate scale, the dodecahedron dihedral angle equals to 120 degrees.

A ray passing through a point $p \in \mathbb{S}^3$ in the tangent direction $v$ is parameterized by $r(t) = \cos t\, p + \sin t\, v$. If $r$ intersects a face of the dodecarehedron we update $p$ and $v$ by the face transformation, which we discuss in more details below.

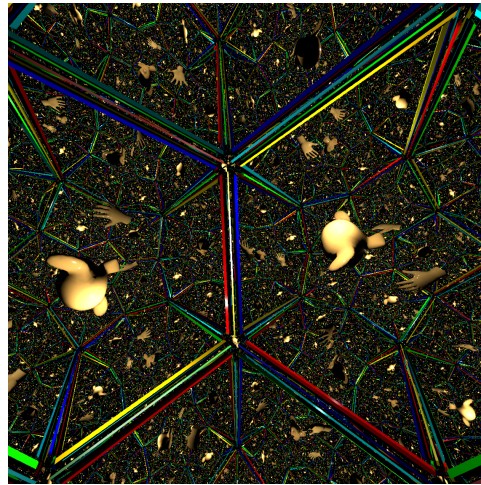

Fig. 4. Immersive visualization of Seifert–Weber dodecahedron space. The space is obtained by identifying, with a rotation of $3\pi/10$, the opposite faces of a special regular dodecahedron embedded in Klein's model of hyperbolic space. We use the dodecahedron to set up our scene: a unique Suzzane with hands and the dodecahedron's edges. The face pairing make the rays that leave a face return, with an additional rotation, from its opposite face, giving rise to many copies of the scene: a tessellation of the hyperbolic space by rotated dodecahedra.

The immersive visualization of Poincaré dodecahedral space is a tessellation of $\mathbb{S}^3$ by 120 dodecahedra. This is one of the 4-dimensional polytopes, known as 120-cell and shown for the first time here.

Figure 5 presents an immersive view of Poincaré dodecahedral space (Example 3). A unique Suzzane with hands and the dodecahedron edges compose the scene. For a better understanding of the spherical geometry, we do not attach Suzzane to the camera. Note that as the distance increases, Suzanne's size first decreases and then begins to increase: there is a large Suzanne upside down at scene background. This image describes the *icosahedron group* acting on the 3-sphere.

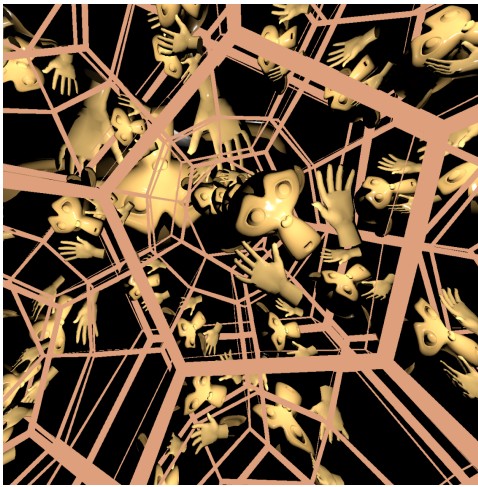

Fig. 5. Immersive visualization in Poincaré dodecahedron space, which is obtained by identifying, with a rotation of $\pi/5$, the opposite faces of a regular dodecahedron embedded in 3-sphere. We use a parameterization of the spherical dodecahedron to set our scene: Suzanne with hands and the dodecahedron's edges. The faces pairing make the rays that leave a face return, with an additional rotation, from its opposite face, giving rise to many copies of the scene: a tessellation of sphere, the 4-dimensional regular polytope known as 120-*cell*, shown here for the first time.

## 5.4 Mirrored Cube

**Example 4** (Mirrored cube). The *mirrored cube $\mathscr{Q}^3$* is an example of an orbifold with the geometric structure modeled by $\mathbb{E}^3$ through a special group of reflection $\Gamma$. Such group is generated by the reflections of the planes $x = \pm 1$, $y = \pm 1$, and $z = \pm 1$ in $\mathbb{E}^3$. The unit cube is the fundamental domain of $\mathscr{Q}^3$. Each time a ray $r$ intersects a face of the fundamental domain of $\mathscr{Q}^3$ it is reflected, creating a polygonal curve in $\mathscr{Q}^3$: exactly what happened with the lights in a mirrored room. Such polygonal curve suspends to ray in $\mathbb{E}^3$, thus we see a tessellation of $\mathbb{E}^3$ by reflected unit cubes when immersed in $\mathscr{Q}^3$.

Figure 6 gives an immersive visualization of the mirrored cube, presented in Example 4. Again, there is a single Suzanne in the scene attached to the camera. The image is the view of a group of reflection acting on the Euclidean space.

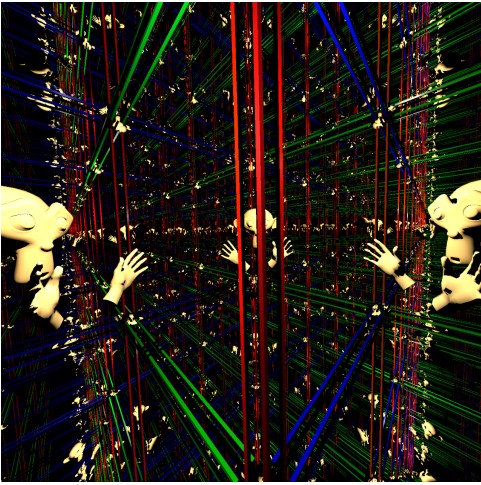

Fig. 6. Immersive visualization of the mirrored cube, obtained by considering the faces of a regular cube to be perfect mirrors. A unique mesh (Suzanne) and the cube's edges provide the scene. The perfect mirrors make the rays iterate, producing the sensation of being inside a cube tessellation of Euclidean space.

## 5.5 Mirrored Dodecahedron

**Example 5** (Mirrored dodecahedron). For an example of an orbifold with a geometric structure modeled by the hyperbolic space, consider the dodecahedron embedded in $\mathbb{H}^3$. Let $\Gamma$ be the group of reflections generated by the dodecahedral faces. With an appropriate scale, the dihedral angle of the dodecahedron reaches 90 degrees. The quotient $\mathbb{H}^3/\Gamma$ is the *mirrored dodecahedral space*. $\Gamma$ tessellates $\mathbb{H}^3$ with dodecahedra, each edge has exactly 4 cells.

Figure 7 illustrates an inside view of the mirrored dodecahedron (Example 5) using the reflection definition in the hyperbolic space. Suzanne model is attached to the camera. The image is the view of the group of reflection acting on the Hyperbolic space.

## 6 ANALYSIS

In this section we present a quantitative and qualitative analysis of the results developed using our framework. This includes computational performance, interactivity and space perception.

### 6.1 Performance

Here we show the experiments to evaluate our algorithm in respect of performance in current VR devices. The hardware setup consists of a computer with a NVIDIA GeForce 2080 Ti for RTX Ray Tracing support and a HTC Vive for VR visualization. The resolution is set to 1512x1680 for each eye, resulting in a total resolution of 3024x3360. A mono version of the algorithm is used as control. Figure 8 shows the results.

Our algorithm achieves performances near 80 fps in high resolution for the Torus, Seifert-Weber and Mirrored Dodecahedra when using 3 or less bounces. This value is near 90 fps, the peak frame rate recommended for VR experiences in Vive, and ensures a smooth experience for users immersed in those spaces.

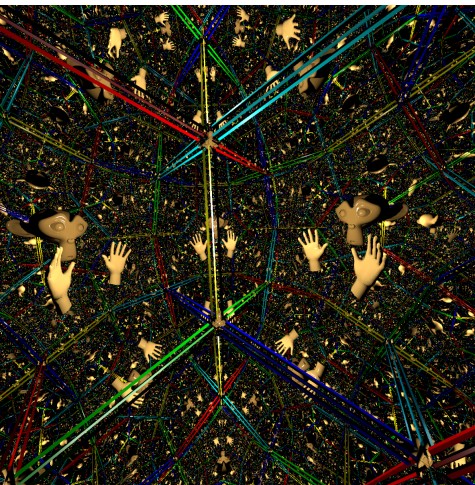

Fig. 7. Immersive visualization of the mirrored dodecahedron. This space is obtained by considering the faces of a hyperbolic regular dodecahedron to be perfect hyperbolic mirrors. A unique mesh (Suzanne) and the dodecahedron's edges provide the scene. The mirrors make the rays iterate in the scene, producing the sensation of being inside a tessellation, by dodecahedra, of the hyperbolic space.

**Performance X Number of ray bounces**

Legend:
- 3 bounces (mono)
- 3 bounces (stereo)
- 5 bounces (mono)
- 5 bounces (stereo)
- 7 bounces (mono)
- 7 bounces (stereo)

Torus: 167, 125, 83, 83, 59, 43
Seifert-Weber Dodecahedron: 167, 83, 77, 63, 42, 29
Mirrored Dodecahedron: 167, 91, 77, 63, 45, 22
Poincaré Sphere: 91, 63, 43, 48, 22, 22

Frame rate (frames per second)

Fig. 8. Performance X Number of ray bounces. The algorithm can generate high resolution stereo images of the spaces, performing up to 80 fps.

## 6.2 Interaction

To give the user a better perception of the torus and the mirrored room, we attach, besides Suzanne's head to the camera, models of the left and right hands to the left and right controls of the HTC Vive (see Figure 6). Thus interacting in the fundamental domain provides a better sense of being immersed in the quotient spaces.

Future works include the motion capture of the user whole body skeleton, using techniques reminiscent from computer vision and artificial intelligence [12] (see Figure 9). This will allow to include in the scene complete avatars of the users, instead of only the head and hands used in the current implementation.

## 6.3 Space Perception

In order to produce a better understanding of the space structure we add the edges of the fundamental domain to the scene. The result gives a perception of a tessellation of the space.

In the above examples, the complete cell structure of the covering space is readily apparent since we explicitly marked the boundary of the fundamental domain, see Figures 3 to 7.

More subtle perception arise if only some static objects are placed in key landmarks of the domain. Moreover, adding a dynamic behavior may give a transient or pulsating character to the space (i.e., with

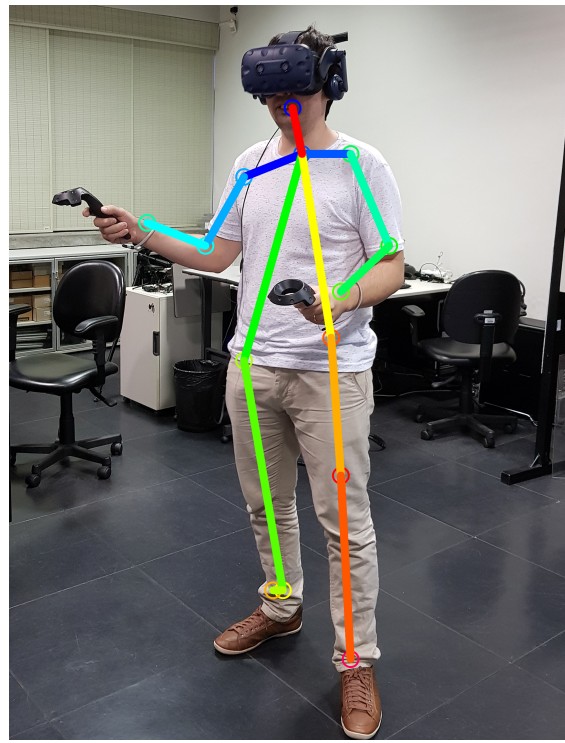

Fig. 9. Pose Detection and Motion Capture: currently Head and Hands are captured using HTC Vive Headset and Controllers; in future implementations the user's pose (indicated by the superimposed skeleton) will be estimated and tracked by the AI method described in [12].

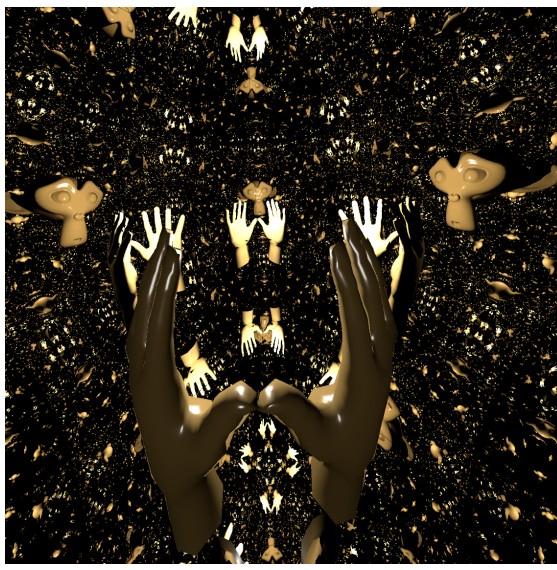

Fig. 10. Inside view of the mirrored dodecahedron. This is the same space depicted in Figure 7 without showing the structure of the fundamental domain.

random or periodic motion, respectively). See Figure 10.

In addition, when the viewer is placed inside an opaque cell with a few openings (e.g., a cube with doors and windows), the perception of an infinite space changes to that of a maze.

Another important ingredient in the understanding of the space structure is the scale, which is related to the fundamental domain volume. In Figures 3 to 5, for example, we are able to see many copies of the fundamental domain, which produce, again, the view of its covering space. However if we consider a fundamental domain sufficiently larger, the user will be able to visualize mostly the immediate surroundings of

the scene restricted to the fundamental domain. This leads us to the philosophical question: *what is the shape of the Universe?* or *could we be living inside a 3D torus?*

## 7 EXTENSIONS AND FUTURE WORK

This work opens many questions related to using virtual reality to visualize abstract spaces. Basically, if in such spaces we are able to compute rays and their intersection with embedded objects (submanifolds, probably), then this opens up many possibilities of exploring the visualization and interaction in such spaces.

### 7.1 Applications

For enhanced user experiences in Non-Euclidean spaces, well-designed *tours* and *games* are desired within these spaces.

For a tour, the virtual user body could be attached to an object, an airplane for example, which is driven by the HTC Vive controllers. This would allow the user to travel among cells of the covering space.

Extending the tour, a game inside these abstract spaces would produce "surreal" experience. Since computing rays is our speciality, a "shooter" game would be a first candidate. Adding some on-off probabilistic rule depending on the cell could give increasing challenge to the player, based on his/her level in the game. Such application is an extension of the two dimensional case presented in [6].

There is also the possibility of many users interacting in a same space. For that, we intend to use the visual motion capture system [12] mentioned in the previous section and the interaction framework of Velho et al. [15].

### 7.2 Illumination, Space and Time Effects

*Fog* is a technique used in rendering to enhance the space perception by letting the object shading to be dependent of its distance from the camera, assuming a participatory medium. Figure 1 (and also Fig. 3) presents the effects caused when applying fog in the space – its left side did not received fog, while the right side received.

For a more artistic application, the scene shading could depend on the cell (thinking that the rays travel in the covering space). Each cell in the covering space has a code $i, j, k \in \mathbb{Z}$, Thus incorporating some on-off probabilistic rule depending on the integers $i, j, k$ could provide a special effect to be exploited by artistic control. Also, adding a delay during the ray tracing would produce a significant time-dependent effect contributing to an artistic space perception.

In future works we intend to investigate other visual effects with different illumination and reflection models. We plan to consider to let the light rays to travel among cells, changing in various ways their contribution of the scene shading.

As mentioned in Section 4.4 we currently adopt only a local illumination model for shading. As a consequence, this prevents the use of global ray tracing effects such as reflection and refraction. Nonetheless, we have already started some experiments to incorporate *path tracing* into our framework. Preliminary results are shown in Figure 11. Unfortunately, the computational performance is not interactive yet and will require many optimizations.

## 8 CONCLUSION

In this paper, we introduced real–time immersive ray tracing visualization algorithms for classical three-dimensional manifolds and orbifolds.

These algorithms are based on the DXR API and are built on top of Falcor, NVIDIA's scientific prototyping framework, which relies on the power of the new generation of RTX GPUs.

Our contribution includes a complete software platform for the visualization of non-Euclidean spaces featuring an efficient and modular architecture that allows the exploration of scenes with both complex geometry and topology, not possible before.

From a theoretical point of view, our framework could be used to investigate abstract phenomena in geometry and topology of low dimension. However our work goes beyond this, it establishes new possibilities for the use of non-Euclidean space in games, art, and in the dissemination of those exotic abstract spaces.

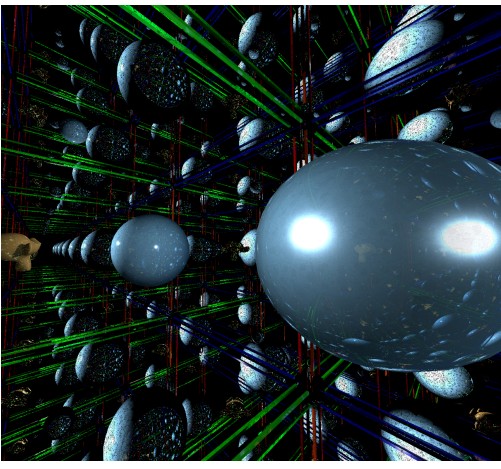

Fig. 11. Global Illumination effects incorporating path tracing to our visualization framework. Inside view of the 3-Torus using the scene of Figure 3 with the addition of an specular sphere, thus the material produces reflections of the ambient.

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
