# OpenReview forum: "Immersive Visualization of the Classical Non-Euclidean Spaces using Real-Time Ray Tracing in VR"
_graphicsinterface.org/Graphics_Interface/2020/Conference — GI 2020_

### Official Review · AnonReviewer1 · 2020-02-10
**Improved Upon Last Submission, Good for Acceptance**

**Rating:** 7
**Confidence:** 4

**Review:**

This paper is a re-submission of the first GI 2020 deadline. The contribution of the paper is exploring visualization of non-Euclidean spaces in VR using RTX GPUs. This work can lead new research and applications into non-Euclidean space visualization. The primary problem of the paper is the same: little technical novelty. The authors compensate by giving detailed explanations on their shader algorithm. Compared to the last submission, the paper structure is improved and some missing details are added. Topology explanation is reduced, increasing readability. Although as a result, some topological concepts like fundamental domains are used without explanation or reference. I would suggest the authors notify users of which references are needed to understand those unexplained concepts.

Some minor points:

- I think the abstract should include more about the paper, like the contribution and experiment, not the NVIDIA's product announcement. I understand Mr. Huang gave a wonderful speech and I too was excited by this. But I would prefer more professionalism in writing research publications.

- Section 6 is missing in Section 1.3 Structure of the Paper.

- An idea for demonstrating the topology: adding animations of objects (e.g. planes) traveling between cells. I think observing how objects move through the space should help user understanding.

- An interesting future work is how to optimize global rendering in those spaces made by recurring cells. Typical rendering methods don't consider such cases and therefore existing algorithms may not help.

Overall I think this is better than the last submission, enough for acceptance into GI after some more polish.

---

### Official Review · AnonReviewer3 · 2020-02-11
**Well-written implementation paper with little technical novelty**

**Rating:** 5
**Confidence:** 4

**Review:**

I reviewed the previous version of this paper submitted to GI's first review cycle. I'm fairly happy with the changes: the unnecessary portions of the math section have been removed, the implementation is described in more detail, and a new supplemental video provides a better view of the system.

However, the central problem I, along with the other reviewers, had pointed out still remains: there is very little technical novelty in the paper. The main effort is the RTX implementation of [3].

I would reiterate to the authors that I appreciate all the changes, but the main concerns have not been addressed. I would encourage looking at the old metareview and following the major recommendations noted in there:
1. User evaluation of the system. Did it help understand the visualized spaces?
2. Exploring novel applications arising from the implementation.

A minor point, with the math subsections removed, is that the term "orbifold" is never defined. I don't think a precise definition is needed, but it should be introduced before the term is used.

---

### Official Review · AnonReviewer2 · 2020-02-14
**the paper is well written, but the technical contribution seems too small and too incremental to me**

**Rating:** 5
**Confidence:** 3

**Review:**

This paper presents a method for visualizing "non Euclidean spaces", based on ray tracing using RTX.

It is sound and well engineered, and the paper is well written and well illustrated. The presentation has been improved since last submission (in particular, the technical level of the introduction is more appropriate).
However, I still believe that the paper itself presents limited novelty: this type of visualization has been proposed already in previous works, and the contribution seems to be limited to the use of the RTX pipeline to obtain faster rendering times than before. This amount of technical contribution is in my opinion below the bar for GI.
I rated the previous submission as "slightly below acceptance level", and my concerns were not really addressed by this revision.

As already written in the previous round of reviews: I think that the presented technique might allow for nice new applications, and demonstrating this might strengthen the paper in a future submission in a Computer Graphics conference. Without new applications (requiring non-trivial technical work), the RTX implementation might not be enough to justify publication in a CG conference.
Another option would be to study more in depth how useful is this type of visualization in practice, in which case the paper could probably be presented in a Scientific Visualization conference.

pros:
- well illustrated, well written
- sound engineering work

cons:
- technical contribution is too small and too incremental (RTX implementation of an existing visualization technique)

---

### Meta-Review · Area_Chair1 · 2020-02-14

**Recommendation:** Accept
**Confidence:** 3

**Metareview:**


The reviewers all appreciated the quality of the writing of the paper, but all pointed out that the technical contribution is extremely limited for the work to be presented at GI. Though the concepts presented in the paper are well known in the CG community, it might still be of interest to the audience at GI.
For these reasons, we recommend to reject the work in its current state, but suggest that a Poster presentation might be adequate.
If the authors choose to resubmit the work in another (Computer Graphics, or Scientific Visualization) conference, the reviewers encourage the authors to increase originality/novelty by, either:
1. Performing a thorough user evaluation of the system. Did it help understand the visualized spaces?
2. Presenting novel, non-trivial, applications arising from the implementation.

Main pros and cons noted by the reviewers:

Pros:
+ System to visualize non-Euclidean spaces in VR with RTX
+ Novel design to show non-Euclidean spaces (by rendering the edges of the fundamental domain with different colors)

Cons:
- Lack of technical novelty

---

### Decision · Program_Chairs · 2020-02-18

Accept